

# Occurrence of seeding multi-layer clouds in the Arctic from ground-based observations

Peggy Achtert[1,2], Torsten Seelig[1], Gabriella Wallentin[3], Luisa Ickes[4], Matthew D. Shupe[5,6], Corinna Hoose[3], and Matthias Tesche[1]

[1]Leipzig Institute for Meteorology, Leipzig University, Leipzig, Germany
[2]Meteorological Observatory Hohenpeißenberg, German Weather Service, Hohenpeißenberg, Germany
[3]Karlsruhe Institute of Technology, Karlsruhe, Germany
[4]Department of Space, Earth and Environment, Chalmers, Gothenburg, Sweden
[5]Cooperative Institute for Research in Environmental Sciences, University of Colorado, Boulder, CO, USA
[6]National Oceanic and Atmospheric Administration, Physical Sciences Laboratory, Boulder, CO, USA

**Correspondence:** Peggy Achtert (peggy.achtert@uni-leipzig.de)

**Abstract.** Studies of Arctic clouds often focus on low-level single-layer clouds (SLCs). Here, we use combined observations of soundings and cloud radar during the MOSAiC, ACSE, and AO2018 research cruises as well as from long-term observations at Ny-Ålesund, Svalbard and Utqiagvik, Alaska to investigate the occurrence of SLCs and multi-layer clouds (MLCs) in the Arctic and to assess the rate of ice-crystal seeding in cold MLCs. MOSAiC observations show cloudy conditions in between 70% and 90% of sounding-radar cases. SLCs dominate during October (90% cases) with otherwise stable rates of around 40%. MLCs are most abundant from November to March (40% to 55% of cases). Seeding occurs in about half to two thirds of the identified MLCs during MOSAiC for which the sub-saturated layer extends between 100 and 1000 m. The seeding rate increases by 5 to 10 percentage points as the assumed size of the falling ice crystals is increased from 100 to 400 $\mu$m. The observations reveal a somewhat higher rate of cloud-free conditions at latitudes south of 84°N. Cloud occurrence during MOSAiC and at Ny-Ålesund in July, when the geographical distance between observations was minimal, shows reasonable agreement. Comparisons of MOSAiC and other research cruises to the central Arctic also indicate consistent occurrence rates of different cloud types despite the likely effect of year-to-year variability. The comparison of data from ship campaigns and land sites suggests that the latter are not a good indicator of cloud occurrence in the high Arctic.

## 1 Introduction

Arctic mixed-phase clouds are widespread, extremely persistent, and most often precipitating (Shupe et al., 2008; Morrison et al., 2012; Mioche et al., 2015; Silber et al., 2021). They have a strong warming impact on the surface and are therefore a critical factor in the Arctic climate system (Cesana et al., 2012; Morrison et al., 2012; Wendisch et al., 2019, 2022). Arctic mixed-phase clouds occur in the temperature range between −38°C and 0°C (Intrieri et al., 2002) where cloud-ice formation takes place via heterogeneous nucleation and relies on the availability of efficient ice nucleating particles (INP). However, Arctic mixed-phase clouds often show an ice crystal number concentration that is orders of magnitude larger than expected from the available number of INP (Rangno and Hobbs, 2001). Aerosol concentrations in general and INP concentrations in particular





are usually low in the clean environment of the central Arctic (Mauritsen et al., 2011; Morrison et al., 2012). The discrepancy between observed ice-crystal number concentration and available INPs is often attributed to secondary ice production (SIP, Gayet et al. 2009; Field et al. 2017; Sotiropoulou et al. 2020) or ice-crystal seeding (Vassel et al., 2019).

The most studied type of Arctic clouds are single-layer low-level stratus or stratocumulus clouds (Morrison et al., 2012). Multi-layer clouds (MLCs) in the Arctic, despite their frequent occurrence, are less well studied. These types of clouds consist of two or more cloud layers that are clearly separated by regions with insufficient water vapour concentration to reach saturation water vapour pressure, i.e., for forming and sustaining a cloud. While the upper layers usually consist of ice or are of mixed phase, lower layers are often predominantly liquid (Achtert et al., 2020; Vüllers et al., 2021). MLCs often occur with multiple
temperature and/or humidity inversions, and as a result of different advection in distinct atmospheric layers (Nygård et al., 2014). Modelling studies (Herman and Goody, 1976; Smith and Kao, 1996; Harrington et al., 1999; Luo et al., 2008; Bulatovic et al., 2023) suggest a variety of crucial processes for the formation and maintenance of MLCs: (i) cloud-top radiative cooling, (ii) destabilization below the upper cloud layer, (iii) surface fluxes, (iv) moistening and latent cooling of lower layers from the evaporation or sublimation of hydrometeors from upper layers, and (v) the seeder-feeder mechanism. In addition, observations
have demonstrated the radiative impact of an overlying cloud layer on cloud-top radiative cooling of a lower cloud layer and, thus, on the strength of in-cloud turbulence (Shupe et al., 2013; Wendisch et al., 2019).

Active remote-sensing observations from ground show that MLCs are by far the most common type of cloud in the Arctic with an occurrence rate close to 50% related to all-sky conditions (Verlinde et al., 2013; Nomokonova et al., 2019; Vassel et al., 2019; Vüllers et al., 2021). However, the expanded view of satellites suggests that Arctic MLC occurrence is particularly
pronounced over the European Arctic (Liu et al., 2012; L'Ecuyer et al., 2019). Ice crystals that precipitate from an upper-level cloud in such a constellation and fall through the cloud-free layer without fully sublimating, can induce glaciation of supercooled clouds in lower layers. This process is referred to as ice-crystal seeding. Depending on its supersaturation with respect to ice or liquid water, the seeded cloud can either start to precipitate and dissipate, or thicken (Korolev et al., 2017).

Observations of mixed-phase clouds in general (Morrison et al., 2009; Korolev et al., 2017) and Arctic mixed-phased clouds
in particular (Avramov and Harrington, 2010; Ovchinnikov et al., 2014) are difficult to reproduce in numerical models (Barrett et al., 2017a, b). The challenge is to properly represent the processes that control the evolution of these clouds, specifically when it comes to the partitioning between the liquid and ice phases within the cloud. In models, phase partitioning is determined, e.g., by the number of available INP as a function of temperature, aerosol abundance and type, the ice habit parameterizations, and the impact of SIP (Sotiropoulou et al., 2020), among others. Young et al. (2019) found that observations of the ice phase
in Antarctic boundary layer clouds can best be reproduced when using a cloud-resolving model and when considering ice-crystal seeding from above. These complex processes have so far only been studied in a small number of realistic or idealized case studies. These modelling studies focus on clouds at Barrow, Alaska (Luo et al., 2008; Morrison et al., 2009; Avramov and Harrington, 2010), renamed to Utqiaġvik in 2016, and at Ny-Ålesund, Svalbard (Schemann and Ebell, 2020). At both locations, the orography, open coastal waters, and adjacent land surface were additional factors of complexity. Modelling
studies of Arctic MLCs over sea ice are still scarce (e.g., Bulatovic et al., 2023; McCusker et al., 2023; Wallentin et al., 2025a) and require benchmark data from atmospheric measurements such as presented here.





The combination of thermodynamic profiles from radiosoundings and cloud-radar observations allows for the identification of MLCs and ice-crystal seeding events (Oue et al., 2016; Vassel et al., 2019). Together with SIP, seeding of a cloud from above can be a major contributor to the high ice crystal number concentrations observed in Arctic low-level mixed-phase clouds (Luo
et al., 2008; Wallentin et al., 2025a). The sounding data are used to identify potentially cloudy and cloud-free layers in which water vapour is saturated and sub-saturated, respectively. In general, this identification depends on (i) whether relative humidity (RH) is considered over water (RHw) or ice (RHi) and (ii) the choice of threshold value.

Collocated cloud radar measurements are then used to confirm if clouds are indeed present in the saturated layers identified from the sounding. Finally, ice-crystal seeding is assumed to occur if the cloud radar data show non-zero reflectivity throughout
a sub-saturated layer (with respect to ice) between two verified cloud layers (i.e., cloud in both sounding in cloud radar measurement) with cloud-top temperature below $0°C$. The presence of such a radar signal indicates that ice crystals from the upper cloud are not fully sublimating while falling through the sub-saturated layer towards the lower cloud. So far, the methodology for identifying ice-crystal seeding in multi-layer mixed-phase clouds from collocated soundings and cloud-radar observations has only been applied to a one-year data set collected at Ny-Ålesund, Svalbard (Vassel et al., 2019) and to the
observations during the Arctic Ocean 2018 expedition Vüllers et al. (2021).

Comprehensive observations of Arctic clouds with cloud radar and soundings have been performed at land sites and during research cruises dating back to the observations during the Surface Heat Budget of the Arctic Ocean (SHEBA) expedition in 1997-1998 (Intrieri et al., 2002; Shupe et al., 2001, 2005, 2006). The next logical step forward in studying Arctic MLCs is therefore to use the wealth of data now available from multiple Arctic sites to systematically investigate the occurrence rate
and seasonality of Arctic MLCs and ice-crystal seeding in Arctic multi-layer mixed-phase clouds. These observation-based findings can then be used to guide detailed studies with cloud-resolving models to gain better understanding of the acting processes.

This work presents an analysis of the occurrence of MLCs and ice-crystal seeding during three research cruises in the central Arctic – including the Multidisciplinary drifting Observatory for the Study of Arctic Climate (MOSAiC) expedition.
These findings are put in contrast to the results of long-term observations at land sites in the European and American Arctic. The paper starts with a discussion of the used data and methods in Section 2. The results are presented in Section 3 and the paper closes with a summary and discussion in Section 4.

## 2 Data and methods

### 2.1 Considered sites and campaigns

While we focus on cloud observations during MOSAiC, we have also considered data from the ACSE and AO2018 ship campaigns as well as from long-term monitoring at Ny-Ålesund, Svalbard and Utqiaġvik, Alaska to put the MOSAiC results into a broader perspective. The locations and time periods of the corresponding data sets are presented in Figure 1. All experiments performed comprehensive cloud observations with soundings, radar, lidar, and microwave radiometer. Here, we only use infor-





mation from the soundings and the cloud radar. The number of daily soundings and the type of cloud radar used to obtain the
different data sets are provided in Table 1.

MOSAiC was based on the German ice breaker Polarstern during its transpolar drift from late 2019 through late 2020, with
a reposition back towards the North Pole in August 2020 (Shupe et al., 2022). Polarstern was moored to an ice floe that slowly
moved from the northern Laptev Sea towards the North Pole and further on to the Greenland Sea. The position of the Polarstern
during the five legs of the one-year ice drift of MOSAiC (Shupe et al., 2022; Silber and Shupe, 2021) is important for data
interpretation as the comparison to other sites and cruises has to account for the change in location and time of the MOSAiC
observations.

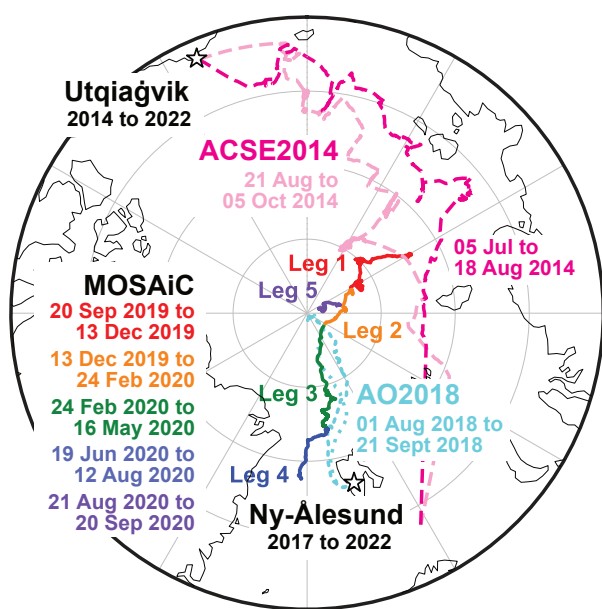

**Figure 1.** Location of the ground stations (black stars) and ice-camp tracks (colours) considered in this study: ACSE (dashed) leg 1 (magenta)
and leg 2 (light pink), AO2018 (dotted, light blue), and MOSAiC (solid) leg 1 (red), leg 2 (orange), leg 3 (green), leg 4 (dark blue), and leg
5 (purple).

Two research cruises with the Swedish ice breaker Oden are used for comparison to MOSAiC findings. The SWERUS-C3
expedition started in Tromsø, Norway on 6 July 2014, crossed the northern parts of the Laptev Sea, the East Siberian Sea, and
the Chukchi Sea to arrive at Utqiaġvik, Alaska. From there, the return leg started on 20 August 2014, crossed the Lomonosov
100  ridge close to the North Pole, and returned to Tromsø on 4 October 2014. The Arctic Cloud in Summer Experiment (ACSE,
Achtert et al. 2020) as the expedition's atmospheric component and included a motion-stabilized W-band Doppler cloud radar
whose measurements were complemented by soundings every 6 h. The Arctic Ocean 2018 (AO2018, Leck et al. 2020; Vüllers
et al. 2021) expedition took place in August and September of 2018 with Oden going to the North Pole for a one-month ice
drift. AO2018 followed up on earlier Swedish research activities related to Arctic climate covering similar times of the year




**Table 1.** Overview of the land stations and ship campaigns considered in this study. Columns list the name of the station or campaign, the type of deployed cloud radar, the start and end times of (considered) observations, the number of daily soundings, and the corresponding (range of) latitudes.

| Station/campaign | Radar type | Start | End | Daily soundings | Latitude |
|---|---|---|---|---|---|
| Ny-Ålesund | 94 GHz, RPG-FMCW | 10.06.2016 | 31.12.2022 | 1 to 4 | 78.92 °N |
| Utqiaġvik | 35 GHz, KAZR | 06.01.2011 | 31.12.2022 | 4 | 71.29 °N |
| ACSE | 35 GHz, W-band | 07.07.2014 | 28.09.2014 | 4 | 71.37–85.22 °N |
| AO2018 | 35 GHz, Mira-35 | 14.08.2018 | 14.09.2018 | 4 | 78.23–89.90 °N |
| MOSAiC | 35 GHz, KAZR | 09.10.2019 | 22.09.2020 | at least 4 | 78.14–88.60 °N |

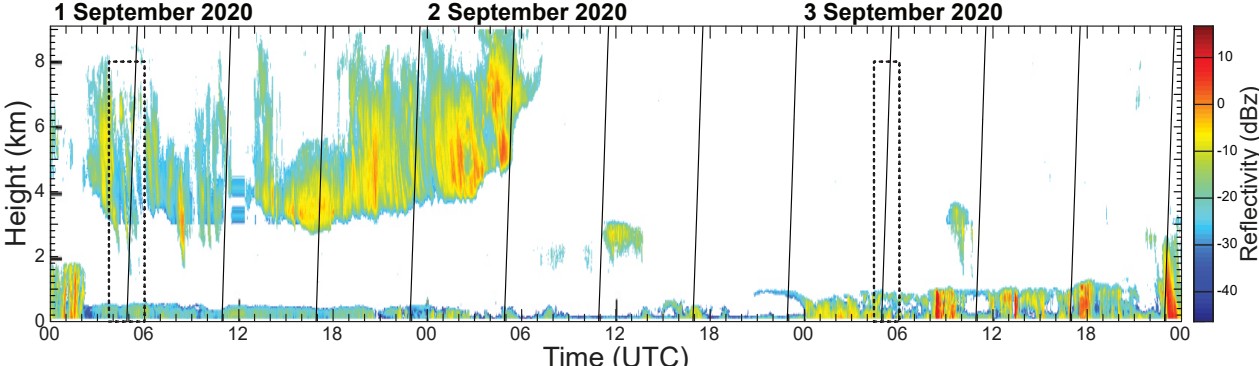

**Figure 2.** Example of cloud occurrence during leg 5 of MOSAiC: radar reflectivity (colour scale) and times of soundings (black lines) between 1 and 4 September 2020. Dashed boxes mark the cases presented in Figures 4 and 5, respectively.

105 and regions dating back to 1996. AO2018 used a 35-GHz scanning Doppler cloud radar and soundings were performed every 6 h (00:00, 06:00, 12:00, 18:00 UTC).

Ship-borne measurements in the central Arctic are usually confined to a few months in summer or early autumn. Two land stations are considered to provide a longer-term perspective of Arctic clouds: Ny-Ålesund (78.9°N, 11.9°E) on Svalbard and Utqiaġvik (71.3°N, 156.8°W) at the North Slope of Alaska. While both sites are further south than any of the shipborne obser-

110 vations, they can provide insight into the annual and inter-annual variation of Arctic cloud occurrence. The site at Ny-Ålesund (Nomokonova et al., 2019; Vassel et al., 2019; Chellini et al., 2023) features a 94-GHz zenith-pointing radar. Soundings are performed once a day at 12:00 UTC but up to four radiosondes per day might be launched during dedicated campaigns. The US Department of Energy's Atmospheric Radiation Measurement facility in Utqiaġvik (North Slope of Alaska site) features a 35-GHz cloud radar and radiosondes are typically launched every 6 h.





## 2.2 Measurement data

We identify the occurrence of cloud layers based on the combination of soundings with coincident cloud-radar measurements (Vassel et al., 2019). An overview of the type of cloud radar and the number of daily soundings at the considered sites is provided in Table 1. Figure 2 shows an example of the spatio-temporal overlap of those observations for a three-day period during MOSAiC that has been used as foundation for the model study of Arctic MLCs in Wallentin et al. (2025a).

The soundings (Maturilli et al., 2022; ARM, 2025a) provide height-resolved observations of pressure (p), temperature (T), and relative humidity over water (RHw) that are used to calculate relative humidity over ice (RHi). The vertical resolution of these measurements is about 5 m based on a 1-s sampling rate.

Radar reflectivity is used to verify the presence of clouds in supersaturated layers identified by radiosondes. The height resolution for the different radars considered here is 3 m at Ny-Ålesund and 30 m during ACSE, AO2018, and MOSAiC, as well as at Utqiaġvik. Radar reflectivity as provided in CloudNet target classification files has been used for ACSE2014 (Achtert et al., 2020), AO2018 (Vüllers et al., 2021), MOSAiC (Engelmann et al., 2021, 2023; Griesche et al., 2024b), and at Ny-Ålesund (Chellini et al., 2023). CloudNet (Illingworth et al., 2007) is a synergetic retrieval that uses observations from soundings, cloud radar, microwave radiometer, and lidar in combination with re-analysis data to identify cloud type and derive cloud properties. CloudNet data feature a unified temporal resolution of 30 s though raw data might be available at higher resolution. ARM data are not systematically processed with the CloudNet algorithm. Therefore, the analysis of cloud presence at Utqiaġvik is based on the original measurement resolution of the cloud radar at that site (ARM, 2025b). At the sites considered here, the lowest radar range gate is located between 80 and 160 m, leading to a near-surface blind zone for confirming cloud occurrence in saturated layers identified in the soundings. This could lead to an underestimation of the occurrence rate of clouds in our analysis compared to studies based on cloud classifiers that combine radar information with observations of ceilometer, lidar, or microwave radiometer for detecting clouds in close proximity to the surface (Illingworth et al., 2007; Shupe et al., 2015).

## 2.3 Detection of cloud occurrence

The detection of cloud occurrence used in this study follows a two-step process that is illustrated in Figure 3 and described in detail in Vassel et al. (2019). Here, we have expanded the output of the method of Vassel et al. (2019) to also provide information on clouds with top temperature warmer than $0°$ as well as SLCs for better comparability to available cloud climatologies.

In the first step, profiles of RHi from a sounding are used to identify the occurrence of layers that are sub-saturated and saturated with respect to ice. Depending on the assumed radiosonde measurement error, threshold values in earlier studies have been set to RHw of 95% (Silber et al., 2020, 2021) and 96% (Silber and Shupe, 2021) for detecting super-cooled liquid clouds and to RHi of 100% for ice-containing clouds (Vassel et al., 2019). In this work, saturation over ice is considered with a threshold value of RHi of 100%. A layer that is saturated with respect to ice is assumed to represent the presence of a cloud layer. Depending on the number of such layers, an observation is classified as (i) cloud-free (no saturated layer with respect to liquid or ice), (ii) single-layer cloud (SLC, one saturated layer with respect to liquid or ice), or (iii) MLC (more than one saturated layer with respect to liquid or ice). The latter cases that also show cloud-top temperatures colder than $0°C$ are further





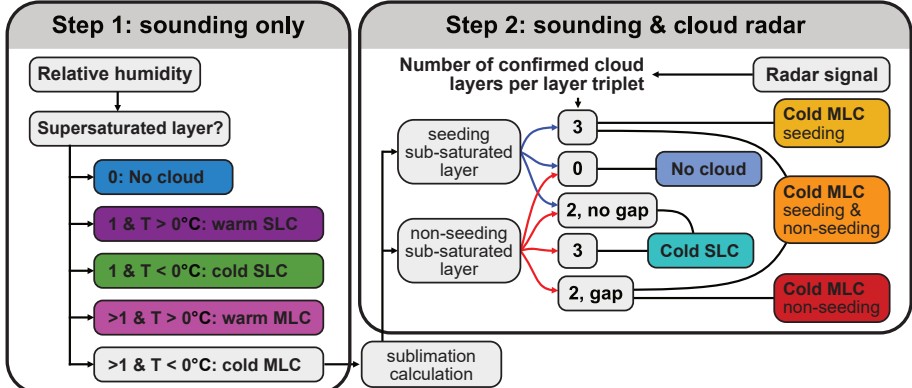

**Figure 3.** Schematic overview of the cloud classification and seeding identification of Vassel et al. (2019) as refined for separating warm (cloud-top temperature above 0°C) and cold (cloud-top temperature below 0°C) single-layer clouds (SLCs) and multi-layer clouds (MLCs), respectively. The two steps relate to using information from soundings (left) and the combination of soundings and cloud-radar measurements (right). The number of cloud layers from 0 to 3 in Step 2 refers to the number of observed cloud layers in the radar measurements for triplets of a sub-saturated layer with saturated layers above and below from Step 1 (see text). The colours refer to different cloudiness, i.e., cloud-free conditions (light and dark blue), warm (purple) and cold (green and turquoise) SLCs, and warm (pink) and cold (yellow, orange, and red) MLCs. The latter are further separated according to the identification of seeding. Note that the colour coding provided here is later used in the presentation of cloud occurrence in Figure 6 onward.

considered as three-layer sets of a sub-saturated layer framed by saturated layers above and below. For each of those layer triplets, sublimation calculations are performed to investigate if an ice crystal released from the bottom of the upper saturated layer sustains a fall through the sub-saturated (cloud-free) layer to arrive at the top of the lower saturated layer. If successful, the case is classified as seeding sub-saturated layer. Cases in which ice crystals fully sublimate while falling through the sub-saturated layer are classified as non-seeding. The sublimation calculations are performed for aggregates, hexagonal plates, rimed columns, and sector plates with radii of 100, 200, and 400 $\mu$m as described in detail in Vassel et al. (2019).

The second step adds cloud radar observations to the analysis to verify that a cloud is indeed present in the saturated layers identified in Step 1. The time period of considered radar observations extends from 30 min prior to the launch of the corresponding sounding until 30 m after it has reached a height of 10 km. Within that time, 50% of the saturated layer need to show radar returns. Missing that threshold leads to reclassification of the layer as cloud-free. If a sufficient fraction of radar returns is observed, their locations are matched with the locations of seeding and non-seeding sub-saturated layers from Step 1. If a sub-saturated layer exists, it forms a layer triplet together with the saturated layers above and below. Situations in which a radar return is present only in either the upper or the lower saturated (cloud) layer are re-classified as SLC (turquoise in Figure 3). Re-classification to SLC also applies if a radar return is detected in either the upper or lower saturated layer and the sub-saturated layer (Vassel et al., 2019, Table 1). Non-seeding sub-saturated layers for which radar returns occur only in the saturated (cloud) layers but not in the sub-saturated (cloud-free) layer of a layer triplet are classified as non-seeding MLCs.





Seeding sub-saturated layers with a radar return observed in all parts of a layer triplet are classified as seeding MLC. The
combination of the latter two cases in a single sounding leads to classification as seeding and non-seeding MLC.

For this work, the method of Vassel et al. (2019), which had been developed for observations at Ny-Ålesund with a single
sounding per day, has been revised to improve applicability to new sites and to make it more user friendly. This includes
(i) restructuring of the code for greater flexibility, (ii) the generalization towards considering multiple soundings per day in
different formats related to common sounding systems, (iii) the output of information related also to warm and cold SLCs
as well as warm MLCs for a more comprehensive investigation of Arctic clouds and better comparability to earlier work,
and (iv) the revision of threshold values related to the geometrical depth of sub-saturated and saturated layers used for cloud
identification. The last item is to reduce the classification of unrealistically shallow cloud layers of only a few meters depth
and increase the reliability of the categorization of SLCs and MLCs. Specifically, saturated (cloud) layers have to show a
depth of 100 m and 500 m below and above 5 km height, respectively, while sub-saturated (cloud-free) layers should be at least
150 m deep. In addition, the threshold of RHi for separating saturated and sub-saturated layers has been set to 100%. This
conservative threshold was selected after sensitivity studies with slightly decreased values produced an unrealistically high
number of supersaturated layers that could not be confirmed in the radar observations. In addition, the scope of this work was
to apply a unified classification methodology without site-specific adaptation.

Finally, soundings during MOSAiC often lead to the detection of a lowermost cloud layer with a top height below 10 m.
This impacts cloud classification towards an increase in the occurrence of MLCs. While the effect might be related to the
occurrence of extremely low-level clouds or fog (Griesche et al., 2024a) it could also be an artefact of releasing the sounding
on the helideck of the RV Polarstern (at an altitude of 12 m) and potential disturbance from the ship's superstructure. As the
occurrence of such low clouds cannot be verified in the radar observations, for which the lowest height bin is at 183 m, the
detection of cloud layers was set to begin above 10 m height in the sounding, i.e., 24 m height above sea level. While the lower
height cut-off was applied to the measurements at all sites and campaigns considered in this paper, differences compared to
including those lower height bins are only found in the MOSAiC observations.

The revisions of the method of Vassel et al. (2019) enable a straightforward and unified application to observations at
sites with comparable instrumental setup and, in principle, also to airborne observations with downward-pointing cloud radar
and dropsondes. This ensures comparability of the findings related to different locations and time periods. In this work, we
are referring to the analysis of a set of coinciding sounding and cloud radar measurements when talking about a case or an
observation.

An example of the application to measurements during MOSAiC is presented for two cases in Figures 4 and 5. The sounding
at 0500 UTC on 1 September 2020 (Figure 4) reveals two sub-saturated layers that are framed by saturated layers with depths
of 0.249 and 3.526 km, respectively. The sublimation calculations for those layers indicate that all considered crystal shapes
and sizes except for 100-$\mu$m hexagonal plates and sector plates survive a fall through the upper (shallower) sub-saturated layer
while not even the fastest-falling 400-$\mu$m aggregates and rimed columns make it through the lower (deeper) one. Consequently,
the two layers are classified as seeding and non-seeding sub-saturated layers, respectively. However, it takes the cloud-radar
observations to corroborate that cloud is present in the different layers for the final identification of cloud type. The upper sub-



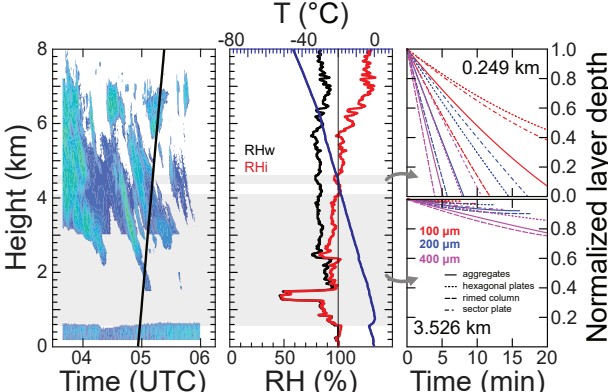

**Figure 4.** Radar reflectivity (colours) and sounding position (black line) (left panel), radiosonde profiles (middle panel) of temperature (blue), relative humidity over water (RHw, black), and relative humidity over ice (RHi, red), and sublimation behaviour over a time period of 20 min after releasing ice crystals from the bottom of the upper cloud layer (right panel) for a sounding launched at 0500 UTC on 1 September 2020. Saturated and sub-saturated layers are marked by white and grey shading, respectively, in the left and middle panels. Colours in the sublimation plots refer to assumed ice-crystal sizes of 100 (red), 200 (blue), and 400 $\mu$m (purple). Line styles mark different ice-crystal habits. The absolute thickness of the sub-saturated layers is given as a number in the respective sublimation plots, for which layer thickness is normalized to vary from zero to unity.

saturated layer at 4.4 to 4.6 km height does indeed have cloud observed throughout its saturated/sub-saturated/saturated layer
triplet. This seeding sub-saturated layer is thus identified as a seeding MLC layer. The cloud-radar observation of a several km deep gap between the cloud below 1 km height and the ice cloud above 3 km height also confirms the lower non-seeding sub-saturated layer as a non-seeding MLC layer. Because both seeding and non-seeding cloud layers are present in this observation, the case is ultimately classified as seeding and non-seeding MLC (see Figure 2).

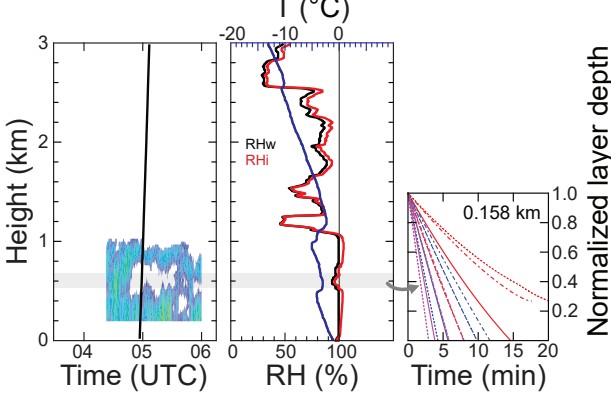

**Figure 5.** Same as Figure 4 but for a sounding launched at 0500 UTC on 3 September 2020.





Figure 5 presents a second case of a low-level seeding MLC related to the sounding at 0500 UTC on 3 September 2020. The
profile of RHi shows a sub-saturated layer around 600 m height with a depth of 158 m. Sublimation calculations indicate that
all but the smallest plates survive a fall through this layer, which is therefore considered as a seeding sub-saturated layer. The
averaged cloud-radar observations for the time period shown in the figure confirm the presence of cloud throughout the layer
triplet and the case is classified as a seeding MLC.

## 3   Results

This section is split in two parts. We start with a discussion of the statistics of Arctic cloudiness from MOSAiC observations
before putting them into the context of other observations during research cruises to the high Arctic and at land sites for
long-term monitoring.

### 3.1   Cloud statistics during MOSAiC

The monthly occurrence rates of different cloud types observed during MOSAiC in Figures 6 show a prevalence of cloudy
conditions throughout the year. The most abundant situations are those in which multiple saturated layers are identified in the
sounding but not confirmed as MLCs in the coinciding radar observation (turquoise). Such unconfirmed cloud layers might
be due to the aerosol limited nature of Arctic cloud formation (Mauritsen et al., 2011) that also impacts the availability of
INP for forming cold clouds. As a consequence, occurrence rates of (warm and cold) SLCs vary between 20% in July and
70% in October. In contrast, MLC occurrence varies between 15% in October and 55% in March. Warm SLCs and MLCs are
found only from June to September and are most relevant during July and August. Within the cold MLC category, 30% to
70% of cases are seeding clouds – with increasing fraction for larger ice crystals that are able to survive a longer fall distance
through the sub-saturated layer. This is generally followed by non-seeding cases that can make up as much as 50% of cold
MLC cases as in July and August. Overall, the combination of seeding and non-seeding cloud layers in the same observation
(orange) are the least abundant type of cloud situation in the MOSAiC data set. This is to be expected as the classification of an
observation as seeding and non-seeding MLC requires the detection of two or more sub-saturated layers (three or more cloud
layers) within one sounding. However, this condition is found in only about one third of all cold MLC observations. Cloud-free
conditions generally persist for less than 30% of observations except for April and July when their rate increases to about 40%.
These numbers agree with Shupe et al. (2022) (their Figure 10e) who provide daily ceilometer-derived cloud fractions during
MOSAiC. April sticks out particularly as it includes periods of close to 90% cloud cover during a warm-air advection event
in the middle of the month followed by the lowest cloud fractions of the campaign of close to 0% shortly afterwards. This
highlights that some of the features observed during MOSAiC are simply due to the synoptic situation for this given year and
may not occur in all years. Motivated by this fact, findings from other available data sets will be presented later in this work.

Barrientos-Velasco et al. (2025) investigate the occurrence rate of cloud-free conditions, SLCs, and MLCs based on com-
bined observations with ceilometer, cloud radar, and microwave radiometer as in Shupe et al. (2015). While they also find high
monthly cloud occurrence rates between 80% and 100%, they resolve less temporal variation in the occurrence of SLCs and





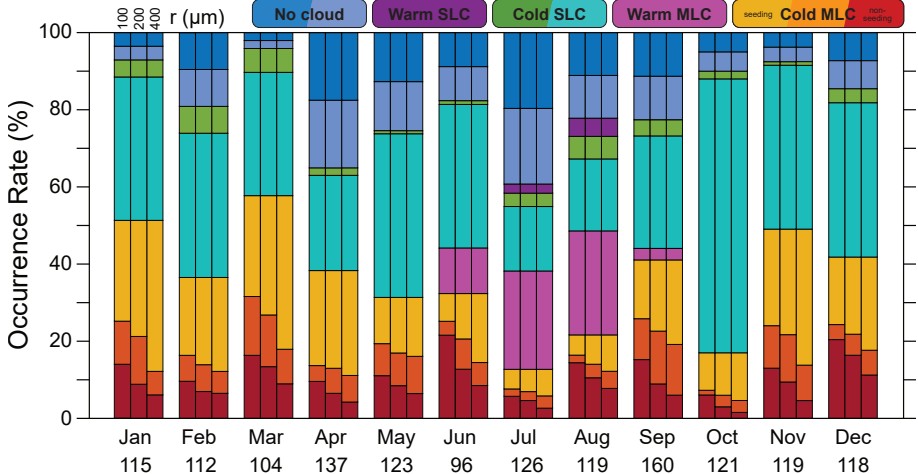

**Figure 6.** Monthly occurrence of cloudiness during MOSAiC according to the cloud classification of Figure 2. Numbers give the sum of considered soundings per month. The three bars per month refer to ice-crystal sizes of 100, 200, and 400 $\mu$m, respectively, assumed in the sublimation calculations for cold MLCs. The colours refer to different cloudiness as defined in Figure 2 from top to bottom of a column: cloud-free conditions (dark and light blue), warm SLCs (purple), cold SLCs (green and turquoise), warm MLCs (magenta), and cold MLCs that are seeding (yellow), seeding and non-seeding (orange), and non-seeding (red). Note the temporal order with respect to month of the year when comparing to other MOSAiC studies that display data from October 2019 to October 2020, e.g., Shupe et al. (2022); Barrientos-Velasco et al. (2025).

MLCs. Figure 6 shows the highest (lowest) MLC occurrence rate of 55% (15%) in March (October), Barrientos-Velasco et al. (2025) (their Figure 2) find values of 55% in August, 25% in November, and around 40% for most other months and as mean value. In addition, their occurrence rate of cloud-free conditions is lower than values of the CERES SYN satellite product they consult as reference. Our findings in Figure 6 are more in line with the satellite reference cloud fraction in Barrientos-Velasco et al. (2025). Finally, our work separates between warm and cold MLCs and also assesses whether the latter are seeding or not.

The interpretation of Figure 6 and the comparison to independent measurements in and around the Arctic is complicated by the fact that MOSAiC observations were not stationary. Figure 7 accounts for the position of the RV Polarstern (see Figure 1) by sorting the findings according to different latitude bands. The figure furthermore provides the annual variation following the definition of seasons based on temperature regimes in Shupe et al. (2022): late autumn from the second half of September 2019 to the end of November 2019, winter from November 2019 to mid-April 2020, spring from mid-April 2020 to the end

of May 2020, summer from June to August 2020, and early autumn from the beginning of September until the end of the expedition. Consequently, such defined seasons show different lengths with spring being the shortest and winter the longest. This is reflected in the number of observations that are considered for the different seasons. In the same way, the time spent in different latitude bands also leads to different sample sizes considered in Figure 7.

The occurrence of different cloud types shows a clear seasonal variation. Late autumn and winter feature the highest occurrence rate of cold MLCs of 30% to 45%, respectively, and the lowest occurrence rate of cloud-free conditions of around 10%.



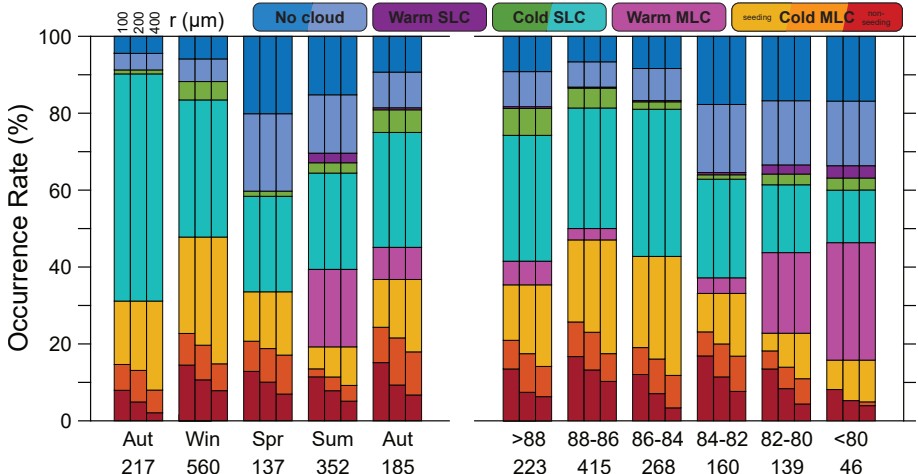

**Figure 7.** Same as Figure 6 but for different seasons (left) and latitude bands (right) to enable comparison to other observations. The two autumn bars refer to observations in autumn 2019 (late autumn) and 2020 (early autumn) at the beginning and end of MOSAiC, respectively.

Cold SLCs make for the remaining 40% to 60%, respectively, during those two seasons. Between about half and two thirds of the cold MLCs during that time are seeding, depending on the assumed size of the falling ice crystals. The contribution of non-seeding MLCs to the total MLC fraction is the lowest during those seasons. Overall, the late autumn and winter observations

are in line with the ones for October to March in Figure 6. Spring shows an increase in cloud-free conditions to around 40%. The occurrence rates of cold SLCs and MLCs are around 30% each, which makes spring the season with the lowest fraction of SLCs. Summer shows the lowest occurrence rates of cold MLCs of around 15% but also features 20% of warm MLCs, in line with the observations during July and August in Figure 6. The fraction of cold MLCs during early autumn is similar to the one in late autumn. However, early autumn also shows a few percent of warm MLCs. Besides spring, cloud-free conditions are

most often found in summer (30%) and early autumn (20%). The occurrence of cold MLCs is reduced during summer as many MLCs occur at warm temperatures and because of weaker synoptic activity during this time period (Rinke et al., 2021; Shupe et al., 2022). Overall, the combination of cold and warm MLCs gives a stable total MLC occurrence rate of 30% to 45% for all seasons, which is in line with satellite observations (L'Ecuyer et al., 2019).

The distinction with respect to latitude band in Figure 7 is introduced to enable a better comparison to the observations from

other expeditions and at land sites discussed below. Note that this presentation is not independent of seasonal variations as a result of the changing position during the ice drift (Figure 1). For instance, the occurrence of warm MLCs is strongly related to observations being performed while heading south during leg 3. As for the seasonally resolved perspective, combined cold and warm MLC occurrence is rather stable at 35% to 45% for different latitude bands. However, the fraction of cold MLCs in general as well as seeding MLCs in particular is higher in the central Arctic (corresponding to September to March, see

Figure 1) compared to latitudes south of 84°N (April to August). This goes along with an increase in cloud-free conditions from about 20% in the central Arctic to about 35% at lower latitudes.





## 3.2 A closer look at cold MLCs

The large contribution of MLCs to the clouds observed during MOSAiC warrants a closer inspection of this cloud scene type and some of the factors that influence the occurrence of seeding.

Figure 8 illustrates that the occurrence of seeding is controlled by the temperature at the bottom of the upper saturated layer (base temperature of the upper cloud), the depth of the sub-saturated layer (the distance between the bottom of the upper saturated layer and the top of the lower saturated layer in a layer triplet), and the size of the falling ice crystals. While the ice-crystal shape also has an effect on fall speed and, thus, distance, it does not compete with the other factors (not shown). For orientation, temperatures slightly below 0°C are typical for clouds during summer while those between -20°C and -30°C

represent clouds during winter. Even lower temperature refers to cases in which the upper cloud is at cirrus level. As expected, seeding is most abundant for shallower sub-saturated layers. It can occur at any cloud-base temperature as long as the fall distance does not exceed 200 m for smaller ice crystals and as much as 700 m for larger ice crystals. While seeding MLCs are found at all upper-cloud base temperatures between -60°C and 0°C, a maximum occurrence rate is visible in the range from -30°C to -15°C and for layer depths between 200 and 400 m (increasing with increasing crystal size). A large number of

seeding MLCs is also found at warmer temperature though this also requires a shallower sub-saturated layer. In some cases, ice crystals larger than 100 μm can also survive fall distances beyond 1.0 km to initiate seeding in the cloud below. In general, lower temperature facilitates longer survival of ice crystals and, thus, larger fall distances. This is a direct consequence of the formulas used in the sublimation calculation (Vassel et al., 2019). Overall, seeding of low-level Arctic clouds, such as in the example of Figure 5, is most likely for upper clouds with top temperature above -25°C that are separated less than 0.2 km from

an underlying cloud (yellow squares in the upper line of Figure 8). The scenarios with increased fall distance of as much as 3.0 km for cloud-top temperature below -40°C are related to seeding within upper-level clouds or cirrus virga, such as in the example of Figure 4.

    In contrast, non-seeding clouds are primarily related to longer fall distance (deeper sub-saturated layers) of more than 200 m for 100-μm crystals to more than 600 m for 400-μm crystals. Nevertheless, there are combinations of fall distance and cloud-

base temperature for small ice crystals (close to the temperature axis in Figure 8) for which a cloud could be either seeding or non-seeding. In line with the larger fall distances of larger ice crystals, an absence of non-seeding clouds is found for shallower sub-saturated layers when increasing ice-crystal size to 400 μm and decreasing cloud-base temperature.

    A clear seasonal variation of the occurrence of sub-saturated layers and ice-crystal seeding is revealed in Figure 9. In general, sub-saturated layers are most abundant during winter and autumn, and rather rare in spring. While spring is the shortest

season with only about two months compared to the nearly four months of winter, the occurrence of sub-saturated layers is disproportionally lower than for other seasons. Most layers show a depth of less than 1.0 km but cases of sub-saturated layers of 3.0 to 4.0 km depth are found in all seasons. Spring and summer show a very similar behaviour regarding the fraction of seeding MLCs despite the larger occurrence rate of sub-saturated layers during summer. On the one hand, no seeding cases are found as the depth of the sub-saturated layer exceeds 1.4 km. On the other hand, the fraction of seeding MLCs increases with

decreasing depth of the sub-saturated layer but stays below 40% for all but the shallowest layers. Autumn shows the highest



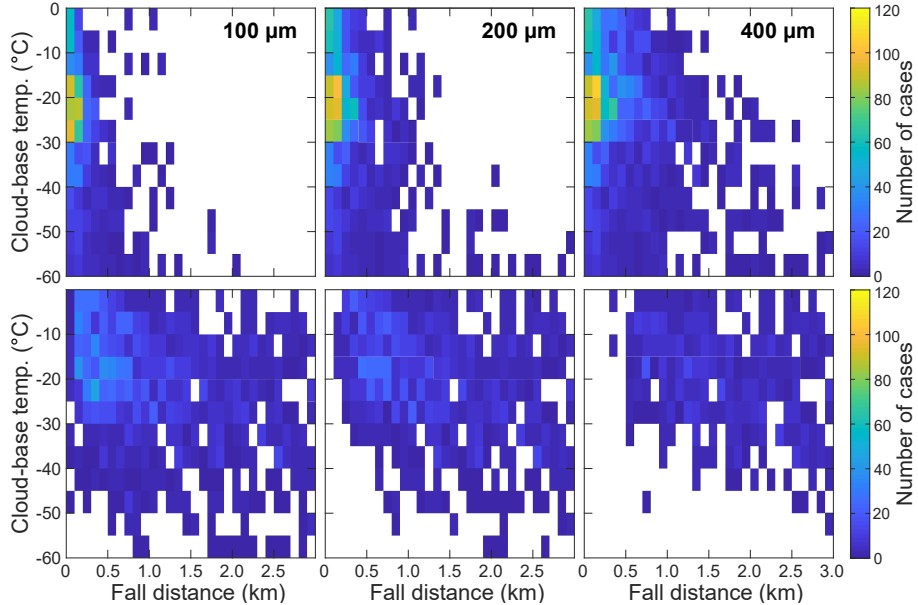

**Figure 8.** Connection between cloud-base temperature of the upper cloud in a layer triplet and fall distance necessary for reaching the lower cloud (depth of the sub-saturated layer) between two cloud layers for seeding (top row) and non-seeding (bottom row) cold MLCs observed during MOSAiC. Columns refer to the assumed size of falling hexagonal ice crystals of 100, 200, and 400 $\mu$m. Cloud geometric information is from the sounding and cloud radar observations. The distinction between seeding and non-seeding conditions results from the output of the sublimation calculation for cases that are identified as cold MLCs as in Figure 1.

occurrence rate of seeding MLCs for sub-saturated layers deeper than 1.0 km. It also shows the lowest seeding rate for the thinnest layers which are also rather rare in terms of absolute number. Observations during winter show the highest numbers of sub-saturated layers independent of their depth. The majority of layers thinner than 1.0 km is associated with seeding MLCs. This fraction generally decreases with increasing layer depth. The high occurrence rate of seeding MLCs at depths of the sub-

saturated layer larger than 1.0 km in autumn and winter is likely related to the presence of cirrus with extended virga as stated in the discussion of Figure 8.

### 3.3 Comparison to measurements around the Arctic

Previous research cruises to the Arctic and long-term measurements at land stations allow for putting the MOSAiC findings in Figures 6 and 7 into a broader perspective. Multiple years of combined soundings and radar observations are available at

Ny-Ålesund and Utqiaġvik as representatives of the European and American Arctic, respectively. At first sight, the monthly occurrence of different cloud types at Ny-Ålesund in Figure 10 shows a very different picture to the MOSAiC observations. Cloud-free conditions dominate with a stable occurrence rate of 40% to 50%. Warm clouds, predominantly warm MLCs, are found between June and September with occurrence rates of 10% to 25% as well as a few cases during October. Cold SLCs and MLCs occur in almost equal amounts except for July when warm MLCs have the strongest impact on the distribution of cloud




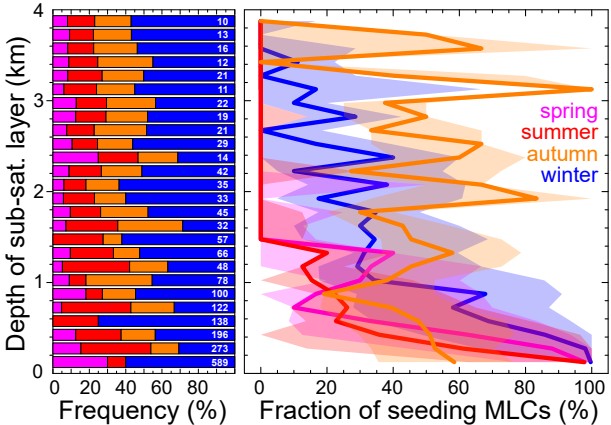

**Figure 9.** Occurrence of cold MLCs with different depths of the sub-saturated layer between upper and lower cloud according to seasonal fraction and total number (left) and fraction of seeding cases (right). The bold lines refer to a crystal size of 200 $\mu$m. The shading marks the range for crystal sizes of 100 $\mu$m (lower seeding fraction) and 400 $\mu$m (higher seeding fraction). Colours mark the different seasons: spring (magenta, 137 cases), summer (red, 352 cases), autumn (orange, 402 cases), and winter (blue, 560 cases).

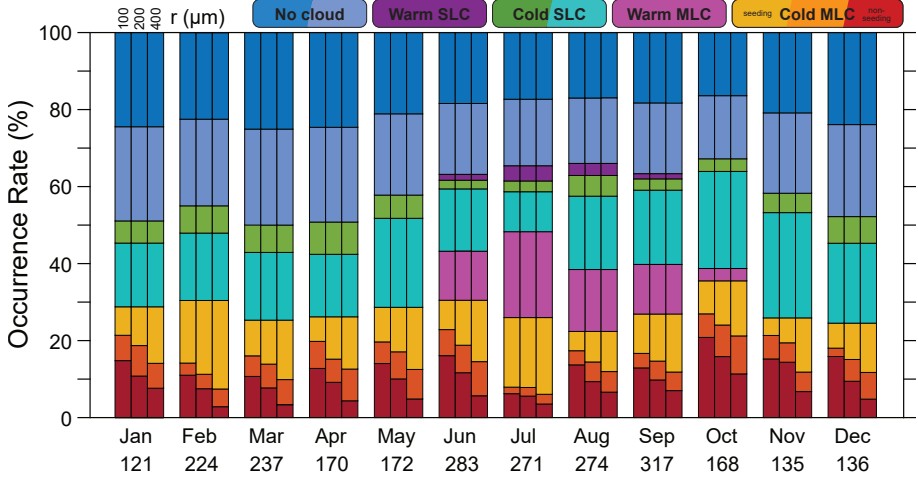

**Figure 10.** Same as Figure 6 but for observations at Ny-Ålesund between 2017 and 2022.

types. Some monthly variation is introduced in the occurrence of seeding and non-seeding MLCs with the latter dominating from October to December and a highest fraction of seeding MLCs in July. A second look shows that the (cold and warm) MLC occurrence rate of around 50% in July is similar to that of around 40% during the same month of MOSAiC in Figure 6. This is a reasonable finding as July observations during MOSAiC were conducted closest to the Ny-Ålesund region (see Figure 1). The MOSAiC observations over sea ice reveal also show a slightly higher fraction of SLCs (20%) compared to those over land at

Ny-Ålesund (10%). Cloud-free conditions make up 35% to 40% of the observations in July at both sites. A similarly consistent



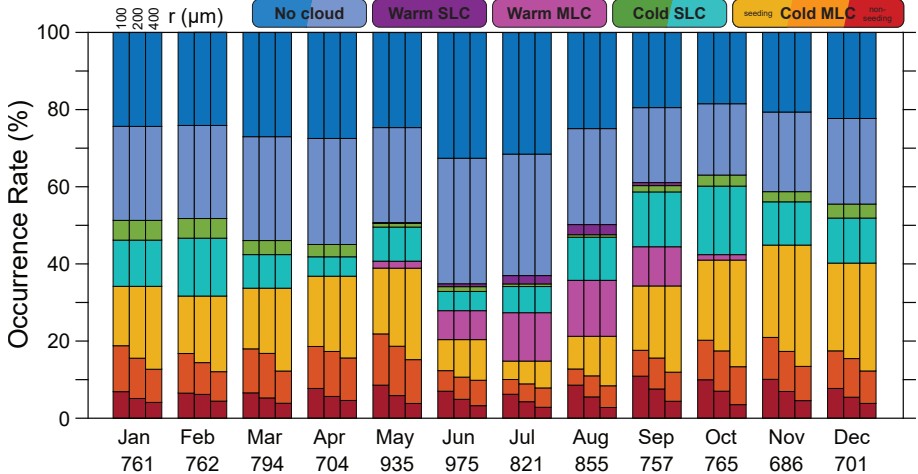

**Figure 11.** Same as Figure 6 but for observations at Utqiaġvik between 2011 and 2022.

picture is found when comparing the summer results for Ny-Ålesund to the MOSAiC observations south of 82°N in Figure 7, i.e., at the latitude band of the land-based observations. This reveals a similar split between cloud-free conditions, SLCs, and MLCs for both MOSAiC and Ny-Ålesund, though with a lower fraction of warm MLCs at the latter site. In contrast to the monthly resolved display in Figure 6, the focus on lower latitudes shows a decrease in the occurrence rate of SLCs and MLC

fractions of around 30% in the MOSAiC data, which are very similar to observations at Ny-Ålesund. Figure 10 also extends the statistics presented in Figure 10 of (Vassel et al., 2019) to multiple years with up to ten-fold increase in considered cases for some months. In contrast to their initial study in which cold SLCs where dominating with occurrence rates of as much as 80%, our larger data set shows basically no seasonality in cloud occurrence except for the warm clouds during summer that have not been part of the presentation in Vassel et al. (2019). A similar qualitative finding regarding the annual variation in

SLC and MLC occurrence has also been presented in Nomokonova et al. (2019). In line with the differences to Vassel et al. (2019), considerable deviations from the occurrence rates in Figure 10 are found when the analysis is restricted to individual years (no shown). This highlights the important role of inter-annual variability and the need for long-term observations when attempting to obtain climatological information on cloud occurrence. In that context, it is also worth noting that the findings in Figure 10 are quite similar to the cloud occurrence rate of 50% to 70% presented in Shupe et al. (2011a) for the time period

from 2002 to 2009.

   As for Ny-Ålesund, observations at Utqiaġvik in Figure 11 show occurrence rates of cloud-free conditions between 60% in summer and 40% to 50% in autumn and winter, respectively. As in case of Ny-Ålesund, considerable inter-annual variability is found also for the time series at Utqiaġvik (not shown). In contrast to the findings at Ny-Ålesund, occurrence rates of SLCs are much lower than that of MLCs. In addition, warm MLCs seem to be less frequent at Utqiaġvik compared to Ny-Ålesund.

We note that the high fraction of cloud-free cases between May and December is not in line with the earlier findings of, e.g., Dong et al. (2010), Shupe et al. (2011a); Shupe (2011b), and Shupe et al. (2015), which consistently show cloud fractions of





70% to 95% during those months. Parts of this difference might be related to the different time period covered in the earlier studies (1998 to 2008 compared to 2011 to 2022). However, it is more likely that differences in the analysis procedure are the reason for the observed inconsistency. The analysis in Shupe et al. (2011a); Shupe (2011b); Shupe et al. (2015) is based on
the combination of lidar, radar, and microwave radiometer with their range of measurement sensitivity to clouds and covered height range. In contrast, we only identify a cloud when a supersaturated layer is found in the sounding data (see Figure 3, Step 1) and the cloud-radar measurement shows radar returns in more that 50% of that layer. On the one hand, this means that the dark blue bars in Figure 10 (no cloud in Step 1) could potentially include cloudy cases that remain undetected because of our conservative RHi detection threshold. We have tested for this by analyzing one year of observations using an RHi threshold
of 96%. However, this analysis indicates that the occurrence rate of cloud-free conditions generally decreases by less than five percentage points when relaxing the RHi threshold. Nevertheless, a dry bias in the considered sounding would lead to an undersampling of saturated layers and, consequently, cloudy conditions, as the methodology used in this study does not consider cloud identification solely based on the radar observations. On the other hand, the requirement for confirming supersaturated layers as cloud in the radar measurement means that our analysis method is blind to clouds below the lowermost radar range
gate that might be identified in the ceilometer or microwave radiometer observations. If such a case coincides with a relatively shallow saturated layer close to the surface, it becomes impossible to meet the 50% threshold of radar returns in the saturated layer required for confirming the layer as cloudy. We have also tested for this effect by omitting all sounding data below 160 m in the retrieval. This lead to a strong increase in cloud-free conditions while SLCs occurrence became almost negligible. Further details regarding the comparison of the cloud-type occurrence rates presented here to cloud fraction as reported in other studies
is provided based on the example of Utqiaġvik observations in 2018 in Appendix A. In summary, we conclude that care should be taken when comparing cloud fraction as obtained here with findings from other cloud classifiers. However, comparing the findings from analysis observations during MOSAiC and at the two land stations with a consistent methodology indicates a clear difference in cloudiness between the central Arctic and the lower Arctic even though this might in part be affected by inter-annual variability.

The long time series at Utqiaġvik with thousands of coinciding soundings and cloud radar observations allows for a comprehensive mapping of where seeding and non-seeding MLCs tend to cluster in terms of cloud-base temperature and fall distance. Figure 12 gives additional confidence in the MOSAiC findings in Figure 8 by confirming the depth of the sub-saturated layer and ice-crystal size as dominating factors for seeding. Seeding is abundant at any cloud-base temperature warmer than -30°C as long as the sub-saturated later is not deeper than 200 m. This depth can increase with larger ice crystals, though seeding
for deeper sub-saturated layers is more pronounced at warmer cloud-base temperatures. Most non-seeding MLC layers seem to occur in a regime of cloud-base temperature and sub-saturated layer depth that is only slightly outside the typical range for seeding MLCs – either at high temperature but longer fall distance or colder temperature. Comparing Figures 12 and 8, we note that non-seeding clouds during MOSAiC occur preferentially at colder temperatures than at Utqiaġvik. This could be related to a larger number of summer clouds with higher cloud-base temperature in the land-based observations or a comparably larger
fraction of seeding cases in mid-level and high clouds in the MOSAiC data set. Testing the latter hypothesis by considering only the lowermost sub-saturated layer does not change the general message of Figures 12 and 8, though.




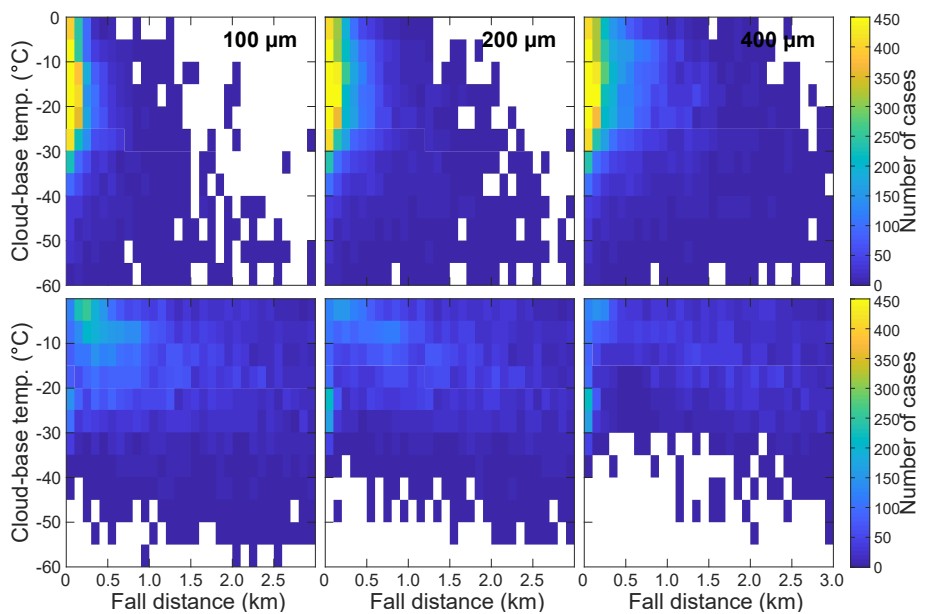

**Figure 12.** Same as Figure 8 but for observations at Utqiaġvik between 2011 and 2022.

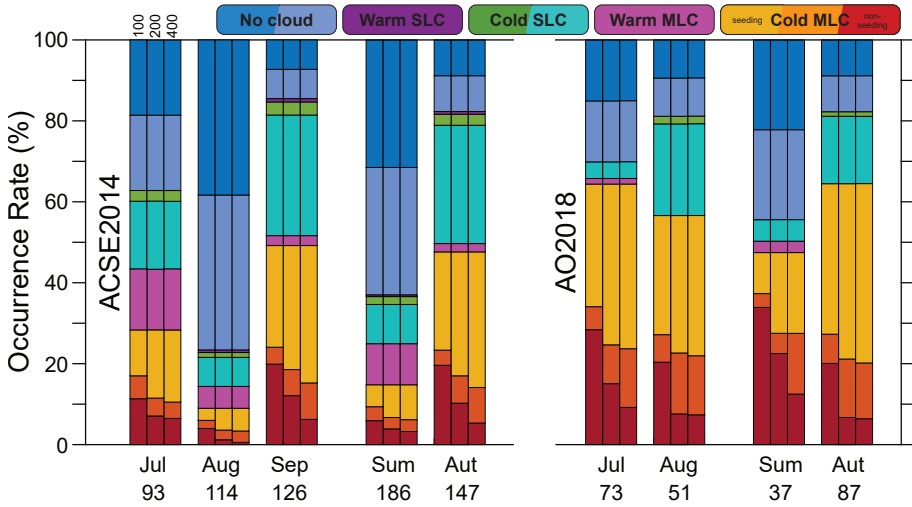

**Figure 13.** Same as Figure 6 but for monthly and seasonal observations during ACSE2014 (left) and AO2018 (right). For ACSE, the transition from summer to autumn is defined by strong temperature decrease that started on 27 August 2014 (Sotiropoulou et al., 2016; Achtert et al., 2020). For AO2018, autumn is set to start with the beginning of ice freeze-up on 23 August 2018 (Vüllers et al., 2021).

The occurrence rates of different cloud conditions during research cruises in the high Arctic with the Swedish icebreaker Oden are presented in Figure 13. These are more of a random sample compared to the observations at long-term land monitoring sites and during the year-long effort of MOSAiC. Observations during AO2018 have been performed the closest to the region



covered during MOSAiC (see Figure 1). MOSAiC findings for July and August show fractions of cloud-free conditions of 40% adn 20%, respectively, that are in reasonable agreement to those during AO2018 of 30% and 20%, respectively. MLCs occurrence rates in the MOSAiC data set of 35% in July and 45% in August are somewhat lower than that of about 50% during AO2018. In addition, AO2018 shows almost now warm MLCs during July compared to 25% during MOSAiC. Another common aspect of both data sets is the relatively low rate of non-seeding MLCs in the European high Arctic during summer.

Observations during ACSE2014 have been performed throughout the Siberian Arctic and enable further connection between observations in the central Arctic and at land stations. July shows a distribution of cloud conditions that is very similar to MOSAiC observations during that month, including the increased occurrence of warm MLCs. Oden was closest to Utqiaġvik during August but the two data sets show large differences in the observations during that month. Cloud-free conditions dominate during ACSE2014 (80%) but make up only 50% at Utqiaġvik which leads to mismatch also for the other cloud types.

This is in part related to the 11-year average presented in Figure 11. The comparison for 2018 in Appendix A highlights the impact of inter-annual variability when comparing results from short-term observations with that from longer time series. In addition, the CloudNet target classification in (Achtert et al., 2020, their Figure 4) indicates a much lower fraction of cloud-free conditions. This inconsistency is likely resulting from the combined effects of low-level saturated layers identified in the sounding and the radar blind zone as outlined in the discussion of the findings at Utqiaġvik as also explored in more depth in

Appendix A.

     ACSE observations during September are the ones closest to MOSAiC in terms of location and show reasonable agreement in cloud occurrence. In both data sets, MLCs contribute to about 40% with a small fraction of warm MLCs, SLCs make for about another 40%, and cloud-free conditions are found only in about 20% of cases. A similar conclusion can be drawn when comparing the statistics for autumn from both MOSAiC (Figure 7) and ACSE (Figure 13). This points towards stabilizing cloud

conditions during autumn freeze-up compared to the higher inter-annual variability during summer that might be modulated by the timing of local sea-ice melt.

     Despite the relatively good match in observational findings, it remains a challenge to draw general conclusions from sporadic research cruises to the high Arctic. The shifting times and locations as well as the long gaps between campaigns complicate the definition of climatological mean cloud conditions from those observations that might be strongly affected by inter-annual

variability. Reanalysis data over longer periods can be used to address this problem in terms of assessing the deviation of atmospheric conditions during a particular campaign from the long-term mean (Rinke et al., 2021).

## 4   Discussion and summary

This study combines observations from radiosondes and cloud radar to quantify cloudiness in the Arctic. The findings from the unique data set collected during MOSAiC are put into the context of observations performed during the earlier research

cruises ACSE2014 and AO2018 to the central Arctic and at the land-based long-term monitoring sites Ny-Ålesund in the European Arctic and Utqiaġvik in the American Arctic. While the comparison of observations at different locations and over





different time periods complicates a straightforward quantification of the results, it is considered as the currently best option for obtaining an observation-based perspective on the occurrence of Arctic MLCs.

MOSAiC observations are dominated by the presence of clouds. Cloud-free conditions make up only about 10% to 30% of observations except for April (35%) and July (40%). The majority of observed clouds from October to February are SLCs with occurrence rates between 40% and 80%. During that time of the year, MLCs make up about 15% to 40% of observations. The balance is shifted towards a more balanced occurrence of MLCs and SLCs between February and September with MLC occurrence rates between 30% and 50%. During most months, about half to two thirds of the identified MLCs are classified as seeding, which suggests the importance of ice-crystal seeding in sustaining the high ice-crystal number concentration in Arctic clouds in relation to the otherwise pristine Arctic environment (Wallentin et al., 2025a). The classification into seeding and non-seeding MLCs is controlled by the depth of the sub-saturated layer between clouds and the assumed size of the falling crystals. Sub-saturated layers that are less than 200 m deep are most likely to support seeding. Seeding is found to also occur in layers as deep as 500 m or more with increasing likelihood for larger ice crystals and at lower temperatures.

When considering the change in position during the ice drift, MOSAiC observations generally show more clouds at higher latitudes and a decrease in cloudiness towards lower latitudes. Specifically, we find SLC and MLC fractions of about 40% each north of 84°N compared to about 25% SLCs and 40% MLCs south of 82°N. A detailed trajectory analysis of the influence of air-mass history on the occurrence of different types of clouds during MOSAiC to complement the data presented here and the analysis in Silber and Shupe (2021) will be the focus of follow-up to this work. The general picture, i.e., cloud-free conditions are more abundant in the lower Arctic than in the central Arctic, is confirmed when considering MOSAiC findings within the context of measurements around the Arctic.

However, the comparison also shows that cloud observations at land sites – while clearly essential for understanding Arctic climate – are not necessarily a benchmark for conditions in the high Arctic. This might be due to differences in, e.g., larger-scale meteorology or the availability of water vapour. Another factor when comparing our findings with those from other studies are methodological differences in the data analysis or variations in the considered data (see also Appendix A). Issues such as a dry bias in the soundings that reduces the detection of saturated layers needed in our methodology for cloud identification or the presence of large parts of a saturated layer at heights below the lowermost radar range gate might explain the differences in cloud occurrence at Utqiaġvik found here and in Dong et al. (2010) or Shupe et al. (2011a); Shupe (2011b). However, applying the sounding-radar cloud-detection methodology to observations during MOSAiC and other Arctic research cruises does in fact lead to cloud fractions of 80% or higher. Together with a sensitivity study in which 96% is used as RHi threshold, this leads us to conclude that the sounding dry bias is not a major error source to the findings presented here. Instead, discrepancies are primarily related to the requirement of filling more than 50% of a saturated layer with radar returns for cloud detection. As a consequence of comparing the MOSAiC findings with observations at land sites, we conclude that the added value of obtaining long-term data sets is countered by the limitation in expanding findings to the wider Arctic region. At the same time, the comparison to observations during earlier research cruises into the high Arctic reveals reasonable agreement to cloud occurrence during MOSAiC in July and August. Finally, using the MOSAiC cloud-type observations presented here as a



benchmark for findings from cloud-resolving modelling provides a pathway for gaining a better insight into the occurrence and properties of SLCs and MLCs throughout the high Arctic (Wallentin et al., 2025b).

The limited number of observations and the lack of a clear baseline of standard conditions for comparison complicate a clear identification of the drivers for the observed differences among sites, which could be related to inter-annual variability in
atmospheric conditions or different timing in the onset of ice melt or freeze up. Satellite observations can potentially provide a reference for long-term cloud conditions in the Arctic. However, current data from spaceborne cloud radar are restricted to latitudes south of 82°N and are biased low at heights below 2.5 km (Mioche et al., 2015; Schirmacher et al., 2023) while passive sensors struggle with the high surface albedo of sea ice and are unable to resolve multiple overlapping cloud layers (Vinjamuri et al., 2023). Despite the achievement of the MOSAiC expedition, this calls for continued efforts to monitor cloudiness in the
high Arctic throughout the year in the form of dedicated research cruises, automated means, and improved analysis methods for satellite observations. Finally, the use of observations of the thermodynamic state of the atmosphere and the occurrence of clouds for initialization and evaluation in targeted modeling studies highlights their importance for improving the quality of numerical weather prediction models in the central Arctic (Tjernström et al., 2020; McCusker et al., 2023).

*Code availability.* The original version of the data analysis code is available at https://github.com/maikenv/Classification_algorithm_of_multilayer_clouds.
The revised version of the data analysis code will be added to the same repository.

*Data availability.* Data considered in this work are available through PANGAEA (https://www.pangaea.de/?q=MOSAIC), the CloudNet data portal (Ny-Ålesund, https://cloudnet.fmi.fi/search/data), the ARM Data Archive (Utqiagvik, https://www.arm.gov/data), and the Bolin Centre Database (ACSE and AO2018, https://bolin.su.se/data/). The MOSAiC CloudNet data used in this study are generated by the Aerosol, Clouds and Trace Gases Research Infrastructure (ACTRIS) and are available from the ACTRIS Data Centre using the following link:
https://hdl.handle.net/21.12132/2.7159000f643348a6.

**Appendix A: How does cloud-type occurrence relate to cloud fraction?**

A closer look at the observations at Utqiaġvik in 2018 is used to address the apparent differences in cloud occurrence (i) at Utqiaġvik reported in Figure 11 and in, e.g., Dong et al. (2010); Shupe et al. (2011a); Shupe (2011b), (ii) at Utqiagvik and during ACSE (Figure 13), and (iii) in Figure 13 and in Figure 4 of Achtert et al. (2020).
The occurrence of different cloud types presented here should not be seen as synonymous with cloud fraction as reported with other multi-instrument cloud classifiers. The latter combine complementary observations with different instruments to identify the presence of clouds. Thermodynamic information from soundings or model fields might then be used to gain a better constraint on cloud phase. In contrast, the method of Vassel et al. (2019) adapted in the work presented here starts with thermodynamic profiles for identifying saturated and sub-saturated air layers and uses radar observations in a second step to
confirm that hydrometeors are indeed present in those layers. No data from other instruments sensitive to cloud occurrence are




considered in the method. As outlined in Section 2.3, more than 50% of a layer has to be filled with radar returns in the hour of observations around a sounding launch for it to be confirmed as cloud or virga. There a several scenarios in which a saturated layer that is indeed filled with cloud might be re-labeled as cloud-free (Figure 3):

– the entire layer is located below the lowermost radar range gate and cannot be confirmed as cloud at all

– the majority of a layer is located below the lowermost radar range gate so that the 50% threshold of radar returns within the layer cannot be met

– a cloud is measured by radar only for a short period within the window around the sounding so that the 50% threshold of radar returns within the layer cannot be met

In addition, a cloud might be present in the radar measurement but in a layer that has not been flagged as saturated in the
sounding data. Such scenarios are not included in our analysis for failing Step 1. As a consequence, our occurrence rates of cloud-free conditions will always be higher than in reports of cloud fraction.

However, we can get an idea to which degree the two parameters are consistent. Figure A1 combines the cloud occurrence at Utqiaġvik for 2018 with cloud fraction as obtained from the ARM Data Base, i.e., including ceilometer and radar data. While occurrence rate and cloud fraction differ in terms of absolute number, they do show similar features such as dips in April, June,
and October. Given the discussion above, one might furthermore conclude that a major part of "cloud by sounding, no cloud by radar" (lighter no-cloud category) does in fact represent cloudy conditions that fail to meet the 50% radar return criterion. If that was the case, the comparison of cloud occurrence and cloud fraction is reconciled to within 15 percentage points during most months of 2018.

The requirement for having 50% radar returns within a saturated layer seems to have the biggest impact for observations
at Utqiaġvik as relatively high cloud fractions are found at most other sites. However, the effect is likely to also affect the observations during ACSE – particularly during August for which Figure 13 gives 20% cloud occurrence while the cloud fraction for the same month is clearly higher in Figure 4 of Achtert et al. (2020) though dominated by clouds with very low cloud-top height.

The comparison of the 11-year Utqiaġvik data set in Figure 11 to the results for 2018 in Figure A1 highlight the role of
inter-annual variability. The long-term data set gives a stable cloud occurrence between 40% and 60% throughout the year with warm clouds from May to October. In contract, the analysis of a single year shows a much larger month-to-month variability in cloud occurrence as well as a wider range in cloud occurrence from 30% in April to almost 80% in December. The 2018 data thus show that it is possible to infer high fractions of cloudiness with our method – another indication that vertical cloud location is key for the apparent inconsistency to observations of cloud fraction.

Using the 2018 data set for comparison to observations during ACSE does not lead to a more consistent picture as that year shows higher cloud occurrence than the multi-year average on top of the issue described in the previous section.





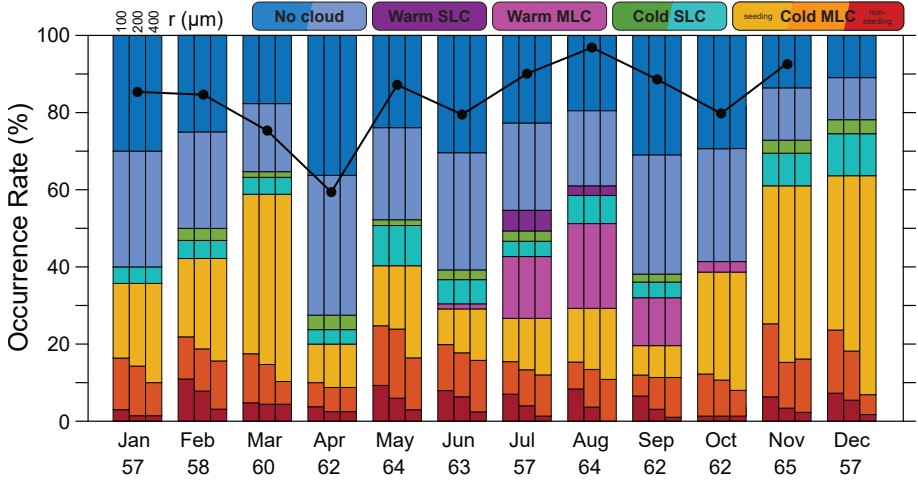

**Figure A1.** Same as Figure 6 but for observations at Utqiaġvik in 2018. In addition, monthly mean cloud fraction as available through the ARM data portal is indicated by the black circles and line.

*Author contributions.* PA and MT conceived the study and acquired the funding. PA, TS, and MT performed the data analysis. All authors contributed to the interpretation of the data and the discussion of the findings. PA and MT prepared the draft manuscript with revisions from all authors.

*Competing interests.* The authors declare no competing interests. LI, and MT are members of the Editorial Board of Atmospheric Chemistry and Physics.

*Acknowledgements.* This work was funded by the German Ministry for Science and Education (BMBF) under grants 03F0891A and 03F0891B. MDS was supported by a Mercator Fellowship with (AC)3, the NOAA Physical Science Laboratory (NA22OAR4320151) and Global Ocean Monitoring and Observing Program (FundRef https://doi.org/10.13039/100018302), and the U.S. Department of En-
ergy (DE-SC0021341). LI was supported by the Chalmers Gender Initiative for Excellence (Genie). We acknowledge ACTRIS and Finnish Meteorological Institute for providing the data set which is available for download from https://cloudnet.fmi.fi. A subset of data used in this study was obtained from the Atmospheric Radiation Measurement (ARM) User Facility, a U.S. Department of Energy Atmospheric Office of Science User Facility Managed by the Biological and Environmental Research Program.



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
