# Peer review of "Occurrence of seeding multi-layer clouds in the Arctic from ground-based observations"

_EGUsphere, 2025_

## Referee Comment (RC2)

**Review of: 'Occurrence of seeding multi-layer clouds in the Arctic from ground-based observations' by Achtert et al.**

**General comments**

In this work, the authors make use of radio soundings and cloud radar measurements from 3 ship campaigns and 2 ground stations. Based on these, they first identify and then classify arctic clouds following a complex, layered classification scheme. They separate between single or multiple layer clouds and focus on their occurrence and the presence of ice crystal seeding in cold MLCs. For these cases they consider the ice crystal size and habit as well as the depth of the subsaturated layer between the cloud layers.

The manuscript is overall very well written. The topic and open question are introduced in an easily understandable manner. The methodology and especially the cloud classification scheme is presented in detail. The results are shown with straight forward plots and explained well. I find the immediate discussion of some results in section 3 very useful.

The topics discussed in this paper, the methodology and findings are in the scope of the journal and the interest of its readers and the scientific community and I would propose this work for publishing after minor revisions and technical corrections.

**Specific comments & questions**

- Figure 3: From the description of the method in the text I understood that all soundings are verified with a radar measurement, but it is not obvious in the graph. If that is the case -although it might escape the scope of this paper-, it would be interesting to know how many SLC and warm MLC were reclassified as no cloud by the radar measurements or to rephrase this, how often there are ice supersaturated regions without clouds.  This would then open many other questions, as for example where, when and why these ISS regions form and what are the dynamics within them.
- Figure 7: As the authors already acknowledge in the text (lines 265 – 266) the latitudinal analysis includes the seasonal bias. Is there an estimate of how strong that is? Maybe with a multiple regression analysis this could be estimated. If it is significant, weighting or performing the latitudinal analysis for each season and averaging could remove the bias. Something similar could also be considered in terms of normalizing the amount of measurements, to support seasonal comparisons.
- Lines 429 – 431: It would we good if the authors could once again mention here, that the latitudinal analysis is also affected by the seasons.
- Lines 436 – 438: Obviously during drifting campaigns measurements are performed only over sea ice, sea ice cover could also play a role in the latitudinal and seasonal variability or even point to a long-term trend.

**Suggestions for technical corrections**

- Figure 1: The black stars are a bit tough to spot. You could consider filling them.
- Line 156: 30 min
- Figure 4 3rd panel: Legend not legible
- Line 214: Figures 6
- Figure 6: 'cloud classification of Figure 3.'
- Line 243: I think seasonal variation would better describe what is presented than annual.

- Line 279: Instead of 'orientation' maybe 'clarification' would be a better word? I immediately though of ice crystal orientation.
- Figure 8: he descriptions 'seeding' and 'non-seeding' could be added to the plot for a faster/ easier understanding of the context.
- Line 324: there seems to be one extra word

---

## Author Comment (AC1)

**General Reply:**

We thank the two Referees for their feedback and specific comments. Please find our point-by-point replies below.

Before getting there, we would like to address a general change that affects Figures 6, 7, 10, 11, 13, and A1 as well as their discussion. During the revisions, we discovered a logical bug in the code for producing the fractional occurrence rates of the different cloud classes shown in those figures. As a result, the occurrence rates of the two *no-cloud* classes were always identical. This issue was also addressed in a specific comment by Referee #2. Re-running the corrected analysis revealed that re-classification of clouds in the sounding that could not be confirmed by the radar observations (no cloud after step 2) actually occurs less often in most data sets than *no cloud by sounding* (no cloud after step 1). We have revised the figures listed above accordingly and changed the corresponding numbers in the text as indicated in the marked-changes file.

We would like to emphasize that those changes do not affect the overall message of the paper. In fact, they lead to a more consistent picture with respect to earlier observation provided in the literatures.

For completeness, the new figures are presented below:

[Figure]

**Figure 6.** Monthly occurrence of cloudiness during MOSAiC according to the cloud classification of Figure 3. Numbers give the sum of considered soundings per month. The three bars per month refer to ice-crystal sizes of 100, 200, and 400 μm, respectively, assumed in the sublimation calculations for cold MLCs. The colours refer to different cloudiness as defined in Figure 2 from top to bottom of a column: cloud-free conditions (dark and light blue), warm SLCs (purple), cold SLCs (green and turquoise), warm MLCs (magenta), and cold MLCs that are seeding (yellow), seeding and non-seeding (orange), and non-seeding (red). Note the temporal order with respect to month of the year when comparing to other MOSAiC studies that display data from October 2019 to October 2020, e.g., *Shupe et al.* (2022); *Barrientos-Velasco et al.* (2025).

[Figure]

**Figure 7.** Same as Figure 6 but for different seasons (left) and latitude bands (right) to enable comparison to other observations. The two autumn bars refer to observations in autumn 2019 (late autumn) and 2020 (early autumn) at the beginning and end of MOSAiC, respectively.

[Figure]

**Figure 10.** Same as Figure 6 but for observations at Ny-Ålesund between 2017 and 2022. In addition, the lower panel shows the standard variation of the occurrence rates of the different cloud classes (same colours as in the bar plot) for the full time period in percentage points.

[Figure]

**Figure 11.** Same as Figure 10 but for observations at Utqiagvik between 2011 and 2022.

[Figure]

**Figure 13.** Same as Figure 6 but for monthly and seasonal observations during ACSE2014 (left) and AO2018 (right). For ACSE, the transition from summer to autumn is defined by strong temperature decrease that started on 27 August 2014 (*Sotiropoulou et al.*, 2016; *Achtert et al.*, 2020). For AO2018, autumn is set to start with the beginning of ice freeze-up on 23 August 2018 (*Vüllers et al.*, 2021).

[Figure]

**Figure A1.** Same as Figure 6 but for observations at Utqiagvik in 2018. In addition, monthly mean cloud fraction as available through the ARM data portal is indicated by the black circles and line.

**Referee #1:**

**Overall impression and rating**

The authors describe the observations and evaluation of seeding events in multilayer clouds during the MOSAiC ship campaign. Although the MOSAiC observations and evaluations are "only" a snapshot of a specific period of time, they are nevertheless very valuable because they are carried out in a region that is rarely observed. In particular, the combination of radiosondes and radar observations in the region is unique for analysis and difficult to achieve with other observation methods. The manuscript reads very well, the illustrations are clear and understandable, and the interpretation is comprehensible. I only have a few minor questions, but no major comments. I can therefore generally recommend the manuscript for publication by ACP.

**Specific comments/questions:**

Temporal evolution of a cloud: You wrote in the appendix that the criterion of 50% radar returns has a major influence on the results. In principle, this goes hand in hand with the temporal development of a cloud. Couldn't this criterion create a bias toward cloud types with certain crystal sizes that seed over longer time scales, i.e., with smaller crystal? This would mean that clouds that seed only for a short period of time or with larger crystal in a smaller part of the cloud would be filtered out. Perhaps you could give your opinion on this.

The discussion the reviewer refers to is about detecting clouds in saturated layers which might fail if less than 50% of the layer also shows radar returns. This effect becomes more important as the depth of the saturated layer increases while the amount of radar returns decreases – as is most likely at larger (cirrus) height levels. The effect is less critical at heights at which seeding might lead to the glaciation of supercooled liquid clouds, which forms the focus of our study. In that context, the criterion is more likely to filter out upper-level clouds than clouds with different seeding behaviour.

Representativeness and inter-annual variability: E.g. Page 16, lines 335-344: Please indicate an example of inter-annual variability (e.g. standard deviation) for example of the cloud free fraction or other fractions of one of the long-term datasets to better get a feeling how large the variability actually is. Even if the inter-annual variability is not the focus of the paper and basically not shown here, it would give the reader an idea of how representative the observations and derived fractions of the MOSAiC campaign actually are even if the location not the same.

The reviewer raises a valid point. We have addressed the issue of interannual variability for the observations at sites with multiple years of measurement by adding the annual variation of the standard variation of the occurrence rates of the different cloud classes as additional panel to Figures 10 and 11. Both figures are provided in our general reply above. The following text has been added to the discussion of the corresponding figures (please see track-changes file for exact position):

*"The standard deviation of the occurrence rates of different cloud classes in Figure 10 illustrates considerable inter-annual variation that is in line with the differences to Vassel et al. (2019). The standard deviation is provided in percentage points to facilitate a straightforward estimate of the possible ranges of occurrence rates for different cloud classes. Using seeding MLCs as an example, we can expect that the values provided in the multi-year average in Figure 10 might be as much as 10 percentage points smaller or larger during October when considering individual years. On the one hand, the annual variation of the standard variation in the occurrence rate of the different cloud classes indicates that August shows comparably little inter-annual variation while statistics for October are most dynamic. On the other hand, some cloud classes, particularly cloud-free conditions, show a strong inter-annual variability for a wide range of months. This highlights the important role*

*of inter-annual variability and the need for long-term observations when attempting to obtain climatological information on cloud occurrence. In that context, it is also worth noting that the cloud occurrence rates of 65% to 80% in Figure 10 are quite similar to the ones presented in Shupe et al. (2011a) for the time period from 2002 to 2009."*

*"Compared to Ny-Ålesund, observations at Utqiagvik in Figure 11 show a different seasonal variation in the occurrence rate of different cloud classes and of the role of seeding MLCs. Despite doubling the length of the time series compared to Ny-Ålesund, considerable inter-annual variability in cloud occurrence is found also at Utqiagvik, particularly for cloud-free conditions and MLCs. Cloud-free conditions vary from 20% in September to 35% during most of January to July. As for Ny-Ålesund, occurrence rates of SLCs of 20% to 30% are generally lower than that of MLCs of 40% to 50%. In addition, warm MLCs seem to show a higher inter-annual variability at Utqiagvik compared to Ny-Ålesund."*

In Ansmann et al. 2025, they reported about "Aged Siberian wildfire smoke polluted the tropopause region over the central Arctic during the entire winter half year of 2019–2020." These heterogeneous formed ice particles are generally larger and sediment to lower altitudes which could also lead to seeding of a second cloud layer. Would that effect your results in comparison to other years with less INP around and how does that fit to your statement on page 10 line 215-218 ?

The absence of radar signals in saturated layers as identified from soundings occurs primarily in the upper troposphere up to around 9 km height. In contrast, the increased INP concentrations reported in Ansmann et al. (2025) were observed at even larger heights. Apart from the absence of aerosols for cloud formation, the uncertainty in the estimated supersaturation with respect to ice does also increase at the low temperature in the upper troposphere and could impact layer detection. We have added this additional item to the discussion and the statement now reads:

*"There is an abundance of situations in which saturated layers are identified in the sounding but not confirmed as SLC (light blue) or MLCs (turquoise) in the coinciding radar observation. Such unconfirmed cloud layers occur primarily in the upper troposphere up to around 9 km height and might be due to the increased uncertainty of the sounding's humidity measurement at low temperatures or the aerosol limited nature of Arctic cloud formation (Mauritsen et al., 2011) that also impacts the availability of INP for forming cold clouds."*

**Technical comments/suggestions:**

Figure 8: I would suggest to add the words "seeding" and "non-sedding" in the label of the colorbar. I.e. "Number of cases" to "Number of seeding cases". Then the reader can see immediately which row belongs to which class.

The labels have been changed as suggested.

Page 16, line337: Please change "(no shown)." to "(not shown)."

Corrected

Page 19, line 388: Please change "now warm " to "no warm"

Corrected

**References**

Ansmann, A., Jimenez, C., Roschke, J., Bühl, J., Ohneiser, K., Engelmann, R., Radenz, M., Griesche, H., Hofer, J., Althausen, D., Knopf, D. A., Dahlke, S., Gaudek, T., Seifert, P., and Wandinger, U.: Impact of

wildfire smoke on Arctic cirrus formation – Part 1: Analysis of MOSAiC 2019–2020 observations, Atmos. Chem. Phys., 25, 4847–4866, https://doi.org/10.5194/acp-25-4847-2025, 2025.

**Referee #2:**

**General comments**

In this work, the authors make use of radio soundings and cloud radar measurements from 3 ship campaigns and 2 ground stations. Based on these, they first identify and then classify arctic clouds following a complex, layered classification scheme. They separate between single or multiple layer clouds and focus on their occurrence and the presence of ice crystal seeding in cold MLCs. For these cases they consider the ice crystal size and habit as well as the depth of the subsaturated layer between the cloud layers.

The manuscript is overall very well written. The topic and open question are introduced in an easily understandable manner. The methodology and especially the cloud classification scheme is presented in detail. The results are shown with straight forward plots and explained well. I find the immediate discussion of some results in section 3 very useful.

The topics discussed in this paper, the methodology and findings are in the scope of the journal and the interest of its readers and the scientific community and I would propose this work for publishing after minor revisions and technical corrections.

**Specific comments & questions**

Figure 3: From the description of the method in the text I understood that all soundings are verified with a radar measurement, but it is not obvious in the graph. If that is the case -although it might escape the scope of this paper-, it would be interesting to know how many SLC and warm MLC were reclassified as no cloud by the radar measurements or to rephrase this, how often there are ice supersaturated regions without clouds. This would then open many other questions, as for example where, when and why these ISS regions form and what are the dynamics within them.

The radar measurements are only considered in case of cold clouds. For warm clouds, RHw is used for identifying the saturated layers and the visual inspection of the combined RH-radar measurements confirms that clouds are always present at RHw = 100%. The issue becomes much trickier for cold clouds. Those are detected using RHi which becomes more uncertain at lower temperatures. We have added a corresponding statement to the beginning of Section 3.1:

*"Such unconfirmed cloud layers occur primarily in the upper troposphere up to around 9 km height and might be due to the increased uncertainty of the sounding's humidity measurement at low temperatures or the aerosol limited nature of Arctic cloud formation (Mauritsen et al., 2011) that also impacts the availability of INP for forming cold clouds."*

We have not looked at the occurrence rate of cloud-free ice supersaturated regions as this is outside the scope of this study. However, we acknowledge that this is an interesting feature that should be inspected in future work.

Figure 7: As the authors already acknowledge in the text (lines 265 – 266) the latitudinal analysis includes the seasonal bias. Is there an estimate of how strong that is? Maybe with a multiple regression analysis this could be estimated. If it is significant, weighting or performing the latitudinal analysis for each season and averaging could remove the bias. Something similar could also be considered in terms of normalizing the amount of measurements, to support seasonal comparisons.

We advise that Figure 7 should be interpreted by consulting Figure 1. The seasonal dependence governs the latitudinal analysis at lower latitudes as all observations below 84°N fall into July and August. In the same way, not all seasons are represented in the other latitude bands. We thought of refining the analysis in Figure 7 along the lines suggested by the Referee. However, we concluded that

the outcome would not contribute much to a better understanding because of the very different sample sizes for different combinations of season and latitude band.

Lines 429 – 431: It would we good if the authors could once again mention here, that the latitudinal analysis is also affected by the seasons.

The beginning of the final sentence of the paragraph has been revised to reflect this comment: *"Even though the MOSAiC observations at different latitudes have been performed during different parts of the year, the general picture…"*

Lines 436 – 438: Obviously during drifting campaigns measurements are performed only over sea ice, sea ice cover could also play a role in the latitudinal and seasonal variability or even point to a long-term trend.

Here, we discuss differences between observations in the high Arctic and at land sites. The issue raised by the Referee seems to be more relevant if we were to compare MOSAiC-like observations during several years.

**Suggestions for technical corrections**

Figure 1: The black stars are a bit tough to spot. You could consider filling them.

Black stars are now filled black.

Line 156: 30 min

Corrected

Figure 4 3rd panel: Legend not legible

The size of the legend has been increased. The legend was also added to Figure 5.

Line 214: Figures 6

Corrected

Figure 6: 'cloud classification of Figure 3.'

Corrected

Line 243: I think seasonal variation would better describe what is presented than annual.

*"Annual"* has been changed to *"seasonal"*.

Line 279: Instead of 'orientation' maybe 'clarification' would be a better word? I immediately though of ice crystal orientation.

Fair point. *"For orientation"* has been replaced by *"For guidance"*.

Figure 8: he descriptions 'seeding' and 'non-seeding' could be added to the plot for a faster/ easier understanding of the context.

The two labels have been added to the title of the colour bar as suggested by Referee #1.

Line 324: there seems to be one extra word

*"reveal"* has been removed